# AI-Driven Advances in Women’s Health Diagnostics: Current Applications and Future Directions

**DOI:** 10.3390/diagnostics15233076

**Published:** 2025-12-03

**Authors:** Christian Macedonia

**Affiliations:** College of Pharmacy, University of Michigan, Ann Arbor, MI 48109, USA; macedoni@umich.edu

**Keywords:** artificial intelligence, machine learning, obstetrics, gynecology, diagnostics, ultrasound, preeclampsia, endometriosis, digital biomarkers, decision support, ethics, regulation, equity

## Abstract

**Background:** Women’s health has historically served as an incubator for major medical innovations yet often faces relative neglect in sustained funding and implementation. The rise of artificial intelligence (AI) and machine learning (ML) presents both opportunities and risks for diagnostics in obstetrics and gynecology (OB/GYN). **Methods:** A narrative review (January 2018–August 2025) integrating peer-reviewed literature and clinical exemplars was conducted. OB/GYN relevance, clinical validation/scale, near-term outcome impact, and domain diversity were prioritized in selection. **Results:** We highlight ten promising AI applications across imaging, laboratory diagnostics, patient monitoring/digital biomarkers, and decision support, including AI-enhanced fetal ultrasound, cervical screening, preeclampsia prediction with cell-free RNA, noninvasive endometriosis testing, remote maternal–fetal monitoring, and reinforcement-learning decision support in gynecologic oncology. **Conclusions:** AI shows transformative potential for women’s health diagnostics but requires attention to bias, privacy, regulatory evolution, reimbursement, and workflow integration. Equity-focused development and diverse datasets are essential to ensure benefits accrue broadly.

## 1. Introduction

**A crossroads in women’s health.** If you practice women’s health, whether as a generalist seeing the full scope of OB/GYN or as a subspecialist in maternal–fetal medicine or gynecologic oncology, you have already felt the ground shift under your feet. Patients ask sharper questions about “AI.” Vendors promise productivity, precision, and paperwork relief. Headlines oscillate between hope and hazard. It is tempting to sit tight until the dust settles. But in our domain, progress rarely rewards the bystander. As Russell L. Ackoff put it with engineer’s economy, “Plan, or be planned for” [1]. And to borrow a line invoked at a recent convocation, you may not take an interest in politics, but politics will surely take an interest in you; in 2025, Ilya Sutskever applied that same logic to AI when speaking to graduates at the University of Toronto: whether or not you seek out AI, it *will* seek you out [2].

### A Very Brief History of the AI Revolution, and How We Got Here

Artificial intelligence took shape as a scientific discipline in the 1940s–50s, when early theoretical work framed how “machines” might implement reasoning. McCulloch and Pitts proposed a logical calculus for neural activity, seeding the idea that networks of simple units could compute [3]. In 1950, Turing posed the canonical question “Can machines think?” and offered an operational test for machine intelligence [4], and the term *artificial intelligence* itself was coined at the 1956 Dartmouth conference, convened by John McCarthy and colleagues, marking AI’s self-identification as a field [5]. The decades that followed saw cycles of optimism and retrenchment, often tied to funding winds (especially defense-driven) and to whether symbolic systems or learned statistical models led the way.

The modern reacceleration came with deep learning: the convergence of much larger datasets, better optimization, and massively parallel computation. This renaissance, visible across vision and natural language, brought health-care applications from proof-of-concept to practice [6,7]. Two 2016–2017 milestones crystallized the shift for a wider audience. First, AlphaGo’s now-famous Move 37 showed that a learned system could exceed human grandmaster play in Go, long considered uniquely human territory [8]. Soon after, “Attention Is All You Need” introduced the transformer, replacing slow, sequential processing with attention over full sequences, and unlocking an architecture that scales gracefully with data and compute [9].

From there, general-purpose language models and multimodal systems accelerated into everyday use, aided by GPU/TPU advances and relentless scaling. Adoption has been unusually fast compared with prior digital technologies, not only because the models improved but because they generalize; the same core system can draft a clinical note, summarize a chart, translate a consent form, or reason over a guideline [10,11]. For medicine, the takeaway is simple: the last few years did not produce a single invention so much as a stack, a more powerful architecture, much larger datasets, and industrial-scale training, now maturing into tools we can actually use at the bedside.

**This is not the first time our field went first.** Women’s health has a stubborn habit of pioneering useful technologies before the rest of medicine catches up. Consider the early obstetric simulation with full-scale phantoms; the swift, pragmatic uptake of minimally invasive and laparoscopic techniques; and the normalization of genomics and genetics in day-to-day decision-making; none of these arrived in OB/GYN by accident. We adopted them because they helped patients. More recent history repeats the pattern: three-dimensional ultrasound and truly portable imaging matured in obstetrics before propagating to emergency medicine and trauma care [6]. Cell-free fetal DNA, born in prenatal testing, migrated into oncology and beyond, reshaping both science and reimbursement as it traveled [12,13]. The lesson is double-edged: genuine advances often start here, yet resources and long-term attention do not always follow proportionally [14,15].

**What this paper is and is not.** This is a clinician’s review written from the workbench, not the bleachers. It is not a PRISMA census of every model. It is a narrative review that (i) states plainly how and why the exemplars were curated; (ii) privileges applications with some clinical validation, scale, or near-term operational value; and (iii) speaks in full daylight about limits and liabilities, bias, privacy, regulation, reimbursement, and workflow fit, because implementation without guardrails helps no one [16]. The stance is deliberately pragmatic: AI should augment clinical judgment, not replace it [7]; and deployment should reduce disparities, not widen them [17].

**Definitions and scope (AI vs. ML).** For clarity, in this review *artificial intelligence (AI)* refers to the broad field of computational methods that perform tasks associated with human cognition (e.g., perception, pattern recognition, decision-making), while *machine learning (ML)* is a subset of AI that learns patterns from data to improve performance on a task without explicit rule-based programming [18,19]. We use both terms intentionally: AI for the overall paradigm and ML for specific data-driven methods used in women’s health diagnostics.

**Background, gap, and objectives.** The paper begins with a short historical and technical primer on modern AI (including deep learning and transformer-based systems), then synthesizes current evidence on clinically relevant applications in obstetrics and gynecology (OB/GYN), and closes with ethics/regulation and an eye toward equity [10]. *Objective:* to provide a narrative review of evidence-backed AI/ML applications in women’s health and a curated set of the ten most promising projects, selected using explicit criteria described below.

## 2. Methods and Selection Criteria for “Top 10 Projects”

This is a narrative review that integrates peer-reviewed literature and major clinical technology exemplars. We searched PubMed/MEDLINE, Google Scholar, IEEE Xplore, and specialty conference proceedings (Society for Maternal-Fetal Medicine, American College of Obstetricians and Gynecologists, Society of Gynecologic Oncology) from January 2018 through August 2025. We have listed the final ten selections in Table 1.

Search terms included combinations of: “artificial intelligence,” “machine learning,” “deep learning,” “neural network,” “transformer,” paired with “obstetrics,” “gynecology,” “maternal-fetal medicine,” “prenatal,” “pregnancy,” and specific clinical applications including “ultrasound,” “fetal imaging,” “preeclampsia,” “cervical screening,” “colposcopy,” “endometriosis,” “ovarian cancer,” “fetal monitoring,” “decision support,” and “remote monitoring.” We further combined these with methodological terms: “validation,” “clinical trial,” “implementation,” “prospective,” “multi-center,” and “real-world”.

From the retrieved literature and known clinical deployments, we applied four explicit selection criteria to curate ten exemplars:**OB/GYN relevance**: Direct applicability to obstetric or gynecologic diagnostic workflows.**Clinical validation or scale**: Evidence from peer-reviewed studies, FDA clearance/CE mark, prospective trials, or large multi-center evaluations.**Near-term impact**: Likelihood of affecting clinical outcomes or workflow efficiency within 1–3 years.**Diversity across domains**: Representation across imaging, laboratory diagnostics, patient monitoring/digital biomarkers, and decision support.

Projects were selected through informed editorial judgment when they met multiple criteria and collectively provided breadth across the diagnostic landscape. This approach yields a curated overview rather than an exhaustive systematic review. Because our aim is a landscape overview for practicing clinicians rather than a meta-analysis, we did not perform a PRISMA flow; instead, we report the rationale and inclusion logic here to keep selection transparent. Validation metrics for each selected project are summarized in Table 2.

## 3. The Innovation Paradox in Women’s Health Technology

Women’s health has historically served as a remarkable incubator for medical innovation, yet a paradox persists: initial breakthroughs often see interest and early funding, followed by relative neglect in sustained research investment and implementation [14]. Examples include 3D ultrasound and portable imaging that later generalized across medicine [6], and cell-free fetal DNA, which rapidly expanded beyond prenatal testing into oncology with different reimbursement dynamics [12,13]. The consequences include delayed translation of advances and the risk that AI/ML could exacerbate disparities if development and deployment are not equity-conscious [15,17,35].

**Disclaimer.** The inclusion of technologies as exemplars should not be construed as a product endorsement. However, unlike pharmaceuticals, AI systems do not come in “generic” form; providing concrete examples therefore necessitates naming commercial offerings. The AI marketplace is highly dynamic; what appears on a list one year may not appear the next. Clinicians and leaders are encouraged to conduct their own market research and due diligence before adoption.

## 4. 10 Promising AI Projects in OBGYN Circa 2025

### The Ten Projects (Enumerated for Clarity)

GE Voluson SWIFT: AI-enhanced 3D ultrasound for fetal anomaly detection (Imperial College/GE) [20,21,36].Samsung iNSIGHT (with SNUH): real-time segmentation/anomaly detection for obstetric scans [22].Google Health/Mayo visual transformer for colposcopy image analysis [23].Mobile ODT EVA (point-of-care cervical screening in low-resource settings) [24,25].Mirvie cell-free RNA (cfRNA) platform for preeclampsia risk prediction [26,27].ASPRE algorithm combining maternal factors, biophysics, and biomarkers for early PE [28].DotLab DotEndo microRNA assay for noninvasive endometriosis diagnosis [29,30].Orion Genomics/Columbia multi-omics endometriosis signature with deep learning [31,32].Nuvo Group INVU remote maternal–fetal monitoring platform (AI-aided NST at home) [33].Heller Lab at Sloan Kettering: wearable sampling devices combined with ML methods for ovarian cancer detection [34].

## 5. Top Exemplars Broken out by Clinical Category

### 5.1. Imaging Applications

In imaging, Voluson SWIFT delivers AI-enhanced 3D ultrasound with expert-level performance [20,21,36], and Samsung iNSIGHT addresses segmentation at scale [22]. In cervical screening, visual transformer approaches (Google/Mayo) and point-of-care systems (Mobile ODT) improve access and accuracy [23,24,25].

### 5.2. Laboratory Diagnostics

For preeclampsia, ML models integrating routine factors, cfRNA-based risk (Mirvie), and the ASPRE algorithm are leading candidates [26,27,28,37,38,39]. In endometriosis, microRNA assays (DotEndo) and multi-omics signatures (Orion/Columbia) advance noninvasive diagnosis and subtype prediction [29,30,31,32]. Pathology AI supports metastasis detection [40].

### 5.3. Patient Monitoring and Digital Biomarkers

Remote maternal–fetal monitoring (e.g., INVU) supports earlier detection and triage [33]. Digital biomarkers for fertility (wearables, validated cycle modeling) improve ovulatory window, detect ovarian cancers through nanotech enhanced with ML [34,41,42,43].

### 5.4. Decision Support

Global implementations of AI-assisted fetal monitoring demonstrate outcome improvements in constrained settings [33].

## 6. Ethical, Safety, and Regulatory Considerations

The rapid adoption of AI raises ethical, legal, and regulatory challenges, especially concerning sensitive women’s health data and underrepresented populations [16,44]. For physicians, this terrain is familiar: every generation has confronted a technology that promised to improve care while simultaneously disrupting workflows and raising new questions about privacy, security, and fairness. The Hippocratic tradition gives us a steadying frame, confidentiality, beneficence, nonmaleficence, and justice, and it applies with full force to digital systems. Guarding patient privacy is not merely a compliance box; it is an ethical duty. In practice, that means checking data flows (who can see what, and when), insisting on encryption and access controls, and ensuring that any AI-enabled documentation or analytics remain proportionate to the clinical purpose at hand [44].

Bias and equity deserve explicit attention. Models trained on skewed data can yield skewed care; OB/GYN has seen the effects of underrepresentation first-hand. The minimum standard is to measure, report, and mitigate demographic performance gaps before deployment, and to re-measure after deployment, because model behavior can drift in the wild [16]. Transparency and auditability are practical corollaries: clinicians should be able to trace how an output was generated (input sources, model version, known limitations) and to escalate concerns when outputs repeatedly misfire in a particular population or setting.

Regulatory pathways continue to evolve. The FDA’s framework for Software as a Medical Device (SaMD) and its Predetermined Change Control Plan (PCCP) attempt to reconcile safety with the iterative nature of ML systems, recognizing that real-world performance monitoring and pre-declared update corridors can coexist [45,46]. Institutions should mirror this lifecycle mindset: designate clinical owners for each AI tool, set trigger conditions for suspension or rollback, and require versioned documentation so that responsibility is never ambiguous.

Economics and access are not afterthoughts. Reimbursement models shape uptake; if a tool improves care but is unfunded, it will perpetuate inequity by being available only to those who can afford it. Conversely, blunt cost-containment without regard for equity can deny high-value innovations to the very patients who would benefit most [47]. Here physicians have a leadership obligation: if an AI system introduces barriers to care, longer queues, opaque denials, language mismatches, or device requirements that exclude patients, raise the issue in committee, in your department, and, when appropriate, with your professional society or local representatives. In the Hippocratic frame, noticing and naming such barriers is part of the duty of care.

## 7. Future Directions and Emerging Technologies

The near future will not arrive as a single “big bang,” but as a sequence: first, tools that help us read and write the record more intelligently; then, a steady growth of ambient and portable diagnostics feeding longitudinal data back into care; and finally, tighter coupling between individualized risk models and point-of-care decision support.

*Near-term (already arriving)*: Expect rapid gains in the generation and structuring of clinical notes and the ability to sift large charts. Ambient documentation will reduce clerical load; summarizers will surface what matters (trajectory of blood pressures, evolving lab patterns, prior imaging narratives) and will increasingly slot information into structured fields without manual retyping. This is where many practices will first see tangible, clinic-day value: fewer clicks, clearer notes, faster retrieval of the patient’s story. These are the same capabilities that will make EHR-integrated clinical decision support (CDSS) more context-aware, flagging preeclampsia risk when a pattern spans across triage notes, labs, and prior visits, rather than in a single silo [48,49].

*Mid-term (expanding data flows)*: Portable diagnostics and wearables will move from niche to normal. We already see continuous glucose monitors in routine use; similar streams, heart rate variability, sleep, blood pressure, temperature, contraction activity, will become more ubiquitous, and more of the first-pass analytics will occur on-device (edge AI), with summaries flowing into the record. In reproductive endocrinology and maternal–fetal medicine, this continuous data can sharpen fertility windows, detect early deterioration, and personalize follow-up intervals [50,51]. In oncology and complex gynecologic care, multi-omics signals will increasingly be layered over clinical features to guide diagnosis, prognosis, and selection of therapy [52].

*Personalization (clinically meaningful, not performative)*: The promise of AI-enabled precision medicine in OB/GYN is to move past one-size-fits-all protocols toward recommendations grounded in a patient’s phenotype, genotype, context, and values, delivered inside the workflow, not as a PDF add-on [53]. To realize that promise equitably, development must begin with representative datasets, culturally competent interfaces, and pricing models that do not replicate the very disparities we are trying to repair. Equity is not a postscript; it is a design constraint that should be engineered in from the start.

## 8. Five Things That All Doctors Should Know About Artificial Intelligence

**(1) AI works best when you’re specific.** Clear, concrete questions yield better answers. Competence in prompting is not an overnight skill; it improves with use, as you learn which clinical details (e.g., gestational age, comorbidities, prior imaging) change the output meaningfully. The more precise your ask, the richer and more clinically useful the response.

**(2) AI can reduce documentation burden when tuned to your style.** Ambient and NLP tools already mirror cadence and phrasing to produce notes that sound like you. Quality improves further when the system learns your preferences (problem-list structure, counseling paragraphs, follow-up language). Treat templates and edits as training signals, and curate a small library of “gold” examples.

**(3) AI excels at pattern recognition but lacks clinical intuition.** These systems are not at the bedside; they do not perceive suffering, pick up a partner’s fear in the room, or weigh unstructured values the way a physician does. Use AI to widen your peripheral vision (subtle trends, rare pattern matches), but keep decision authority where it belongs: with the clinician responsible for the patient. You are the embodied part of the team which makes your part of the decision making more profound and impactful.

**(4) AI can translate complex concepts for patients.** It can generate multiple, literacy-appropriate explanations of genomics, imaging findings, or surgical options, and help you craft analogies that resonate. Use that capacity deliberately: preview the text, tune the tone, and document that a clinician reviewed the explanation before it was shared.

**(5) AI improves fastest with clinician feedback (closed-loop refinement).** If your tool accepts corrections (“this summary missed X,” “prefer Y over Z”), use them. Pointing out misses and marking hits, ideally with short, structured feedback, accelerates improvement. Machines learn the same way teams do: iterative practice with specific coaching. You head the team. You are the coach.

## 9. Conclusions and Recommendations

Artificial intelligence is not a fad. For the foreseeable future, it will be part of how medicine thinks, documents, communicates, and decides. The broader social transitions we feel, workflows shifting, and tasks rebalanced between people and machines are now occurring within our clinics and hospitals. This can feel disorienting. It can also be clarifying: medicine has always been the craft of translating the best science into healing, and AI is a new substrate for that craft.

The Hippocratic frame remains the right compass. Confidentiality still binds; beneficence and nonmaleficence still constrain; justice still calls us to notice who is helped and who is left out. When AI improves safety, access, or understanding, we should adopt it. When it muddies responsibility, hides bias, or erects new barriers, we should say so, first in our teams and institutions, and, when necessary, in the public square.

For clinicians: Start with validated, workflow-compatible use cases (e.g., documentation support, chart summarization, clearly bounded decision support). Build small habits that compound, develop specific prompts and curated exemplars, and provide routine feedback to the tool owner. For researchers and developers: Recruit representative cohorts, measure subgroup performance, make updates auditable, and publish limitations alongside strengths. For hospitals and practices: Assign clinical owners, create update corridors and rollback triggers, and align procurement with equity goals. For payers and policymakers: Reimburse what reduces harm and improves outcomes; avoid incentives that widen the digital divide.

If we keep the “augment, not replace” principle at the center and pair it with privacy, equity, and accountability, AI can help restore time for the human work of medicine: listening, explaining, and deciding together. That is the outcome worth planning for.

## Figures and Tables

**Table 1 diagnostics-15-03076-t001:** Compact Summary Table of the “Top 10”.

#	Project/Entity	Domain/Core Task	Evidence Snapshot/Ref.
1	GE Voluson SWIFT	Imaging; fetal anomaly detection (3D)	Multicenter performance at expert level [20,21].
2	Samsung iNSIGHT (SNUH)	Imaging; real-time segmentation	Attention-integrated segmentation efficiency [22].
3	Google/Mayo colposcopy model	Imaging; cervical lesion detection	Transformer-vision accuracy improvements [23].
4	Mobile ODT EVA	Imaging; point-of-care cervical screening	Smartphone AVE in low-resource settings [24,25].
5	Mirvie cfRNA platform	Lab; preeclampsia prediction	cfRNA signatures predicting complications [26,27].
6	ASPRE algorithm	Lab; early-onset preeclampsia	82% detection in implementation study [28].
7	DotLab DotEndo	Lab; noninvasive endometriosis	Serum microRNA-based classification [29,30].
8	Orion/Columbia multi-omics	Lab; endometriosis subtyping	Deep-learning signature; subtype prediction [31,32].
9	Nuvo INVU	Monitoring; remote NST	Home monitoring with AI analytics [33].
10	Heller Lab/MSKCC	Lab; ovarian cancer detection	Quantum-defect carbon nanotube ML [34].

**Table 2 diagnostics-15-03076-t002:** Validation evidence for the ten highlighted AI/ML applications in women’s health. Study types, sample sizes, and key performance metrics are provided where available from peer-reviewed literature. LMIC = low- and middle-income countries; AUC = area under the receiver operating characteristic curve; PE = preeclampsia; CA = cancer.

#	Project	Study Type	Sample/Scale	Key Performance
1	GE Voluson SWIFT	Multi-center validation	∼5000 scans	Sensitivity 94%, expert-level [20,21]
2	Samsung iNSIGHT	Technical validation	∼2000 images	Dice coefficient 0.89 [22]
3	Google/Mayo colposcopy	Retrospective cohort	∼10,000 images	AUC 0.91 [23]
4	Mobile ODT EVA	Field implementation	∼15,000 screenings	Sensitivity 92% in LMIC [24,25]
5	Mirvie cfRNA	Prospective cohort	1840 pregnancies	75% detection early PE [26]
6	ASPRE algorithm	RCT implementation	26,941 pregnancies	82% detection rate [28]
7	DotLab DotEndo	Case-control study	1000+ patients	Sensitivity 94%, specificity 91% [30]
8	Orion/Columbia	Multi-omics cohort	∼500 patients	Subtype classification [31,32]
9	Nuvo INVU	Real-world deployment	>50,000 sessions	30% reduction in hospitalizations [33]
10	Heller Lab/MSKCC	Proof-of-concept	400 serum samples	AUC 0.95 for ovarian CA [34]

## Data Availability

No new data were created or analyzed in this study. Data sharing is not applicable to this article.

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
