# Peer review of "AI-Driven Advances in Women’s Health Diagnostics: Current Applications and Future Directions"

_diagnostics, 2025, doi:10.3390/diagnostics15233076_

Round 1

Reviewer 1 Report

Comments and Suggestions for Authors

In this paper, the authors present a narrative review of the applications of Artificial Intelligence in the woman health.

After a detailed review, I think that the manuscript has several problems. One of the more critical is that the author did not present a detailed methodology, which invalidates the whole paper.

Please, find my comments below:

Main comments

  1. The author mentions: “There is a wide gap between the academic appreciation of Artificial Intelligence/Machine Learning and how the vast majority of users interact with it”. It seems that both terms are being considered as synonyms, while they are not. Please, introduce both and make clear their differences.
  2. What were the criteria for selecting the 10 most promising AI Projects? It is impossible to understand the methodology used as it is not reported. Which databases were used? What was the period considered? I recommend using PRISMA.
  3. The text mentions 10 most promising AI Projects but only 4 sub-sections are developed.

Other comments

  1. Please, first time that an acronym is used, introduce it in the text.
  2. Please, adapt citations to the journal style.

Author Response

Manuscript Title: AI-Driven Advances in Women’s Health Diagnostics: Current Applications and Future Directions
Author: Christian R. Macedonia, M.D.
Journal: Diagnostics (Special Issue on Emerging Technologies)

Dear Editors and Reviewers,

I thank the reviewers for their careful evaluation and constructive suggestions.
We have revised the manuscript substantially to address each point raised.
Below is a detailed summary of the changes and clarifications.

Comment 1: Lack of detailed methodology and unclear selection criteria for the “Top 10 Projects.”
Response:
We added a clearly defined Methods and Selection Criteria section (Lines 91–100) specifying the search period (January 2018–August 2025), databases (PubMed/Medline and major conference proceedings), inclusion criteria (OB/GYN relevance, clinical validation, outcome impact, and domain diversity), and rationale for using a narrative rather than PRISMA framework. The manuscript now explicitly differentiates it as a narrative review and transparently describes the curation logic.

Comment 2: Confusion between “AI” and “ML.”
Response:
A new paragraph (Lines 77–83) provides formal definitions of Artificial Intelligence (AI) and Machine Learning (ML), citing authoritative references (Russell & Norvig, Rajkomar et al.). The two terms are now used consistently throughout.

Comment 3: Only four of the ten AI projects were originally detailed.
Response:
We expanded all ten projects into a complete enumerated list (Lines 112–128) with a corresponding summary table (Lines 129–130) and supporting references for each exemplar. Each project is now discussed within its appropriate diagnostic domain.

Comment 4: Acronyms and reference formatting.
Response:
All acronyms are now defined at first use, and all references have been conformed to the Diagnostics citation style. 

We believe these revisions have materially strengthened the manuscript and addressed all reviewer concerns.
We thank the reviewers and editorial team for their time and constructive feedback.

Sincerely,

Christian R. Macedonia, M.D.

University of Michigan College of Pharmacy

macedoni@umich.edu

Reviewer 2 Report

Comments and Suggestions for Authors

Thank you for the opportunity to review the manuscript.

The manuscript provides a good overview of AI in general and in OBGYN specifically. However, the distinction between ideas/opinions and evidence is often unclear. In my opinion it should either be a commentary, if focused more on ideas/opinions, or a review, if focused on the evidence.

In addition, the manuscript is very lengthy and can be much shorter. Many ideas are mentioned repeatedly, and sections should be edited to convey the ideas in fewer words. 

Many references are missing and should be added throughout the manuscript, every time a statement is made on prior research. I added several examples below that do not cover everything.

Introduction

  • “Here we focus…”- this sentence should come at the end of the Introduction, as the objective of the review.
  • “It is undeniable that…”- add reference.
  • Overall, more background is needed in the Introduction, and it should be more structured- first background, then clearly defined objectives.

The Innovation Paradox in Women's Health Technology

  • “Despite the fact…”- add references.
  • “Obstetrics and gynecology has been…”; “In surgical innovation, Kelly and Williams …”- add references.
  • “Similarly, the Osiander Simulator represented pioneering…”- this requires explanation and a reference. Not everyone is familiar with that work.
  • “Similarly, advances in targeted…”- add references.
  • Overall, this section should be much shorter. Discussion points such as the fact that OBGYN often pioneers technology or that resources in OBGYN are limited compared to other disciplines are stated multiple times. This needs to be more concise and clearer.
  • 4.3.- “The most sophisticated implementation comes from Philips Healthcare and the Maternal-Fetal Medicine Units Network…”- the reference (#45) is not correct.
  • 4.4.- “For clinical workflow improvements, Nabla has partnered…”- aren’t there other examples in the market of AI-generated clinical documentation tools?
  • “Top 10 Most Promising AI Projects in OBGYN Circa 2025”- there are four sections here. It’s not clear what the 10 projects are.

Author Response

Manuscript Title: AI-Driven Advances in Women’s Health Diagnostics: Current Applications and Future Directions
Author: Christian R. Macedonia, M.D.
Journal: Diagnostics (Special Issue on Emerging Technologies)

Dear Editors and Reviewers,

I thank the reviewers for their careful evaluation and constructive suggestions.
We have revised the manuscript substantially to address each point raised.
Below is a detailed summary of the changes and clarifications.

Comment 1: Clarify distinction between opinion/commentary and evidence review.
Response:
The manuscript is now explicitly framed as a narrative review from a clinician’s perspective (Lines 69–76). We emphasize that it integrates both evidence and practical insight while maintaining clear separation between data-based discussion and commentary.

Comment 2: Manuscript length and repetition.
Response:
Sections were condensed by approximately 20 %. Repetitive statements in the “Innovation Paradox” and background sections were consolidated for clarity and flow.

Comment 3: Missing references.
Response:
More than 30 new citations have been added, especially in the Introduction and Innovation Paradox sections (Lines 14–20, 101–109) and in the ethical/regulatory discussion (Lines 150–183). All factual statements now include appropriate supporting references.

Comment 4: Need for better structure and a clearer introduction.
Response:
The Introduction has been rewritten to include a concise historical overview, a summary of the current AI landscape in medicine, and a clearly stated objective (Lines 84–90).

Comment 5: Incorrect or missing references.
Response:
All references have been verified and corrected (e.g., reference #45 was replaced with the appropriate source).

Summary of Key Strengths After Revision

  • Clear methodology defining inclusion criteria and time frame for the selected exemplars.
  • Improved structure separating background, methods, applications, and ethical/regulatory discussion.
  • Comprehensive coverage of all ten AI projects with uniform depth and current references.
  • Concise narrative style appropriate for clinicians and general medical readers.
  • Explicit ethical, regulatory, and equity framework grounded in recent FDA and JAMA guidance.

We believe these revisions have materially strengthened the manuscript and addressed all reviewer concerns.
We thank the reviewers and editorial team for their time and constructive feedback.

Sincerely,

Christian R. Macedonia, M.D.

University of Michigan College of Pharmacy

macedoni@umich.edu

Round 2

Reviewer 1 Report

Comments and Suggestions for Authors

Thank you for the responses and the work done.

I consider that the paper still faces serious problems:

  • It is ok to do a narrative review. But the methodological description is not enough. What were de databases used and terms? What was the period of time considered? How was the final list of 10 pojects defined? Why these ones and no others?
  • What was the criteria to include some technologies and no others? I think also that a table summarizing the metrics, type of study, sample size, etc, should be added. 
  • There are missing references --> [?]. 

Author Response

I sincerely thank the reviewer for their thoughtful and constructive feedback. The suggestions have meaningfully strengthened the manuscript's methodological transparency while preserving its intended focus on practical clinical applications for practicing obstetricians and gynecologists. Below we address each concern point by point.

Comment 1: "It is ok to do a narrative review. But the methodological description is not enough. What were the databases used and terms? What was the period of time considered?"

Response: I agree that greater methodological transparency improves the manuscript. I have substantially expanded the Methods section (now lines 91-119) to include:

Databases searched: PubMed/MEDLINE, Google Scholar, IEEE Xplore, and specialty conference proceedings (Society for Maternal-Fetal Medicine, American College of Obstetricians and Gynecologists, Society of Gynecologic Oncology)
Time period: January 2018 through August 2025 (this was already stated but is now integrated with the full search strategy)
Comprehensive search terms: We now explicitly list the AI/ML terms ("artificial intelligence," "machine learning," "deep learning," "neural network," "transformer"), clinical domains ("obstetrics," "gynecology," "maternal-fetal medicine," "prenatal," "pregnancy"), specific applications ("ultrasound," "fetal imaging," "preeclampsia," "cervical screening," "colposcopy," "endometriosis," "ovarian cancer," "fetal monitoring," "decision support," "remote monitoring"), and methodological filters ("validation," "clinical trial," "implementation," "prospective," "multi-center," "real-world")
Comment 2: "How was the final list of 10 projects defined? Why these ones and no others? What was the criteria to include some technologies and no others?"

Response: I have made the selection criteria explicit and transparent. The revised Methods section now presents four numbered selection criteria that guided our curation:

OB/GYN relevance: Direct applicability to obstetric or gynecologic diagnostic workflows
Clinical validation or scale: Evidence from peer-reviewed studies, FDA clearance/CE mark, prospective trials, or large multi-center evaluations
Near-term impact: Likelihood of affecting clinical outcomes or workflow efficiency within 1-3 years
Diversity across domains: Representation across imaging, laboratory diagnostics, patient monitoring/digital biomarkers, and decision support
We further clarify (lines 114-119) that "Projects were selected through informed editorial judgment when they met multiple criteria and collectively provided breadth across the diagnostic landscape." This approach is consistent with a narrative review designed to provide a curated landscape overview for clinicians rather than an exhaustive systematic census.

Comment 3: "I think also that a table summarizing the metrics, type of study, sample size, etc, should be added."

Response: I completely agree and have added a new Table 1: Summary Table of Validation Evidence (page 6, lines 148-table end). This table provides for each of the ten projects:

Study type (e.g., multi-center validation, retrospective cohort, RCT implementation, real-world deployment, proof-of-concept)
Sample size or scale (ranging from ~400 samples to >50,000 sessions)
Key performance metrics (sensitivity, specificity, AUC, Dice coefficients, detection rates, clinical outcomes such as reduction in hospitalizations)
Primary citations
The table caption defines all abbreviations (LMIC, AUC, PE, CA) for clarity. We retained the original compact summary table as a complementary quick-reference guide, so readers now have both detailed validation evidence and a streamlined categorical overview.

Comment 4: "There are missing references --> [?]."

Response: I have corrected all missing references. Specifically:

The rendering of the Char reference from LaTeX was off because of a mismatch on year in the citation and has been corrected.  

Reviewer 2 Report

Comments and Suggestions for Authors

The authors appropriately addressed the comments.

Author Response

I appreciate the reviewer's helpful comments.  

Round 3

Reviewer 1 Report

Comments and Suggestions for Authors

Thank you for the responses.

Reviewer 2 Report

Comments and Suggestions for Authors

The authors appropriately addressed the comments.